# Denoising Pretrained Black-box Models via Amplitude-Guided Phase Realignment

**Hongliang Ni**  *uqhni@uq.edu.au*
*School of Electrical Engineering and Computer Science*
*University of Queensland*
*ARC Training Centre for Information Resilience*

**Tong Chen**  *tong.chen@uq.edu.au*
*School of Electrical Engineering and Computer Science*
*University of Queensland*
*ARC Training Centre for Information Resilience*

**Shazia Sadiq**  *shazia@eecs.uq.edu.au*
*School of Electrical Engineering and Computer Science*
*University of Queensland*
*ARC Training Centre for Information Resilience*

**Gianluca Demartini**  *g.demartini@uq.edu.au*
*School of Electrical Engineering and Computer Science*
*University of Queensland*
*ARC Training Centre for Information Resilience*

**Reviewed on OpenReview:** *https://openreview.net/forum?id=526fwttJiK*

## Abstract

Pre-trained models tend to inherit noisy label information from their training datasets, internalising it as biased knowledge. While learning with label noise has been explored, existing approaches rarely address the mitigation of biased knowledge embedded in pre-trained representations introduced by noisy labels. Moreover, existing denoising methods invariably rely on modifying training datasets or models to improve downstream task performance. However, we observe a growing trend in which both pre-trained models and their training datasets are scaling up significantly and becoming increasingly inaccessible, making modifications ever more infeasible. In this paper, we propose a black-box biased knowledge mitigation method called "Lorem", which leverages feature frequency amplitudes to guide phase correction on pre-trained representations, without access to training data or model parameters. We first present empirical evidence that, across different noise levels, the phase components of pre-trained representations are more sensitive to noisy labels than the amplitude components, while discriminative information for classification is primarily encoded in the amplitude. Moreover, we find that the impact of noisy labels on amplitude is global, leading to a gradual loss of discriminative information. Therefore, corrective strategies must be adaptive across the entire frequency spectrum rather than limited to the high-frequency components. Inspired by this observation, we design a method that leverages the amplitude residual to realign phase, thereby removing biased knowledge from pre-trained representations. Experiments on a variety of popular pre-trained vision and language models suggest that, even with a simple linear classifier, our method can enhance downstream performance across a range of in-domain and out-of-domain tasks.

# 1 Introduction

As public expectations for the performance of pre-trained models continue to rise, to meet these expectations, pre-trained models increasingly rely on web-crawled data and large-scale annotations, which makes these large-scale pre-training datasets inevitably affected by noisy labels (Piktus et al., 2023). A growing concern is that noisy labels affect the learned weights of pre-trained models, leading to biased knowledge being encoded in their representations. For example, when benign tumours are mislabeled as malignant during pre-training, the model may overemphasise irrelevant pixel-level artefacts. This biased knowledge affects the generalisation and transferability of pre-trained models in downstream tasks, and addressing the problem by simply fine-tuning the representations with downstream data is often suboptimal. Since noisy labels alter the feature space, these distortions constrain the optimisation capacity of fine-tuning, especially when the downstream dataset is small or domain-shifted.

Existing research in learning with noisy labels has mainly focused on the training phase, with mainstream methods generally categorised into noise modelling (Van Rooyen et al., 2015; Han et al., 2018a; Van Rooyen & Williamson, 2018; Yao et al., 2019) and model-based techniques (Reed et al., 2014; Liu & Tao, 2015; Thulasidasan et al., 2019; Lyu & Tsang, 2019; Yu et al., 2019; Han et al., 2020). These methods assume access to the model's training data, architecture, and parameters, allowing them to mitigate the effects of label noise to some extent. However, such strategies are difficult to apply to improving the performance of large-scale pre-trained models on downstream tasks. Due to their size, re-training pre-trained models is often computationally infeasible—even when noisy labels are identified. Moreover, both the datasets and models are not always publicly available. Therefore, it remains an open problem to design effective strategies for mitigating the influence of noisy labels in pre-trained models in a black-box setting, where neither the original data nor model parameters are accessible.

In this paper, we explore a frequency-based approach to mitigating biased knowledge in pre-trained representations, leveraging the Fourier transform—a global transformation that decomposes representations into frequency components. This decomposition allows us to disentangle amplitude (global energy distribution) and phase (structural layout), enabling us to identify and correct frequency-specific distortions introduced by noisy supervision, thereby mitigating biased knowledge. We begin by presenting an empirical analysis, where amplitude and phase signals obtained under different levels of noisy supervision are used as predictive features, and the performance of classifiers built on these features is compared across downstream tasks. Our analysis shows that both amplitude and phase are affected as the noise level increases. However, classifiers built on amplitude signals consistently outperform those using phase, suggesting that phase should be the primary target of correction, as restoring distorted phase components is key to recovering the semantic integrity of pre-trained representations and improving their quality for downstream tasks. To further investigate, we examined how noisy labels affect amplitude and found that as the noise level increases, the magnitude, structural patterns, and distribution of amplitude have significant global changes, rather than being limited to the high-frequency components emphasised in prior work (Chen et al., 2021). These fluctuations imply that amplitude encodes label-related global patterns that are progressively distorted by noisy supervision. Thus, while amplitude itself is affected, its variations provide a reliable signal of where and how noise alters the feature space, making it a natural guide for phase realignment, where amplitude residuals are used to realign phase components, thereby mitigating biased knowledge in pre-trained representations.

Motivated by these findings, we propose "Lorem", a lightweight black-box framework designed to improve the generalisation ability of pre-trained representations. We first transform pre-trained features into a label-guided feature space, while preserving their original amplitude information. We then compute the amplitude residuals—defined as the difference between the amplitudes in the original feature space and those in the new feature space—and use them to guide the correction of the phase components. The aligned phase, along with the new amplitude, is combined via the inverse Fourier transform to generate enhanced representations, aiming to remove biased knowledge and support more reliable transfer to downstream tasks.

The main contributions of this paper are as follows:

- We provide an empirical analysis of amplitude and phase components under varying levels of label noise, demonstrating their differences in sensitivity to noisy labels and their relevance to discrimi-

native information. Furthermore, subsequent experiments reveal that label noise globally changes both the structural patterns and the magnitude of amplitude. This highlights the potential of using amplitude to guide the correction of phase for mitigating biased knowledge in pre-trained representations.

- We propose a lightweight black-box method, named "Lorem", which uses amplitude residuals to guide the adjustment of phase. The enhanced features are then reconstructed via the inverse Fourier transform, allowing for improved generalisation and transferability of pre-trained models with minimal computational cost.

- We conduct extensive experiments on a variety of popular pre-trained vision and language models, evaluating both in-domain (ID) and out-of-domain (OOD) downstream tasks. Our method is compared against strong baselines and widely-used fine-tuning techniques, consistently demonstrating superior performance.

## 2 Problem Formulation

We assume the pre-training dataset is defined as $D = \{x_i, y_i\}_{i=1}^N$, where it consists of inputs $x_i \sim \mathcal{X}$ and supervision $y_i \sim \mathcal{Y}$, and the dataset size is $N$. The inputs and corresponding supervision can take any form. For example, in an image classification task, the inputs are images and the supervision are class labels (Russakovsky et al., 2015; He et al., 2016); in a sentence-pairing task, both the inputs and outputs are sentences (Cer et al., 2017; Wang et al., 2018). We further assume that the pre-training dataset contains noisy labels, such as data samples annotated with incorrect labels, which is common in large-scale pre-training datasets collected using web crawlers and similar techniques.

Thus, we define the noisy dataset as $\tilde{D} = \{x_i, \tilde{y}_i\}_{i=1}^N$, which we assume is inaccessible. The pre-trained model can be abstractly viewed as a combination of a feature extractor and a projection head. The pre-trained representations $\mathbf{F}$ are extracted from the feature extractor $f_\phi$, and then mapped into the target domain using the projection head $g_\theta$. Similarly, the parameters of the feature extractor $f_\phi$ are also assumed to be inaccessible. Since the training dataset contains noise, the extracted pre-trained representations may not accurately reflect the semantic content of the samples. Therefore, our goal is to repair the pre-trained representations to enhance the performance of pre-trained models on downstream tasks, without requiring re-training. To achieve this, we assume access to a downstream dataset $\mathcal{D}' = \{x_i, y_i\}_{i=1}^K$, which is used for optimization. Unless otherwise stated, we consider $\mathcal{D}'$ to be clean (i.e., without additional label noise).

## 3 Analysis of frequency component robustness across noise levels

Given current amplitude–phase recombination approaches for learning with noisy labels (Huang et al., 2023; Chen et al., 2021), we are interested in investigating the downstream performance of the frequency components—amplitude and phase—of pre-trained representations obtained from models under different levels of noisy supervision. In this analysis, we use five noisy pre-trained ResNet-50 (He et al., 2016) models from Chen et al. (2024), each trained with different levels of label noise, each trained with a different label noise ratio (0%, 5%, 10%, 20%, and 30%, where 0% denotes the clean dataset) using CNN backbones, to extract features from four image datasets: CIFAR-10, CIFAR-100 (Krizhevsky et al., 2009), Food 101 (Bossard et al., 2014) and ImageNet Val (Russakovsky et al., 2015) . We then apply the Fourier transform to obtain their amplitude and phase components, and compare their performance in linear probing.

As shown in Figure 1, We have several observations:

(1) The classifiers using amplitude signals consistently outperform those using phase features in downstream classification tasks. This can be attributed to the pre-trained model's inherent tendency to encode label-related semantic information into the amplitude. The most visually obvious differences between categories—such as colour intensity, edge thickness, and texture density—are directly associated with the amplitude. During training, the model tends to prioritise memorising such easily accessible information to converge more quickly, thereby embedding most of the discriminative information into the amplitude.

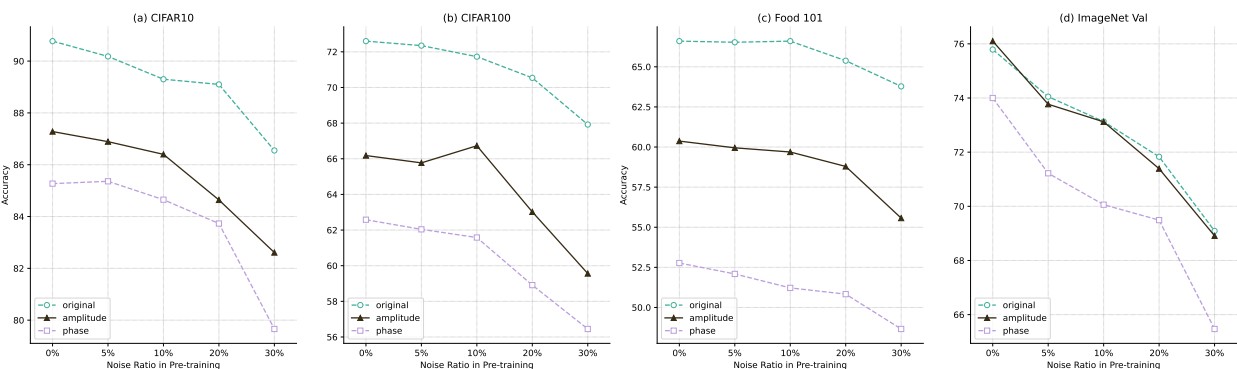

Figure 1: Classification accuracy on CIFAR-10 (a), CIFAR-100 (b), Food 101 (c), and ImageNet Val (d) using original pre-trained representations, their amplitude, and their phase under varying noise ratios in pre-training.

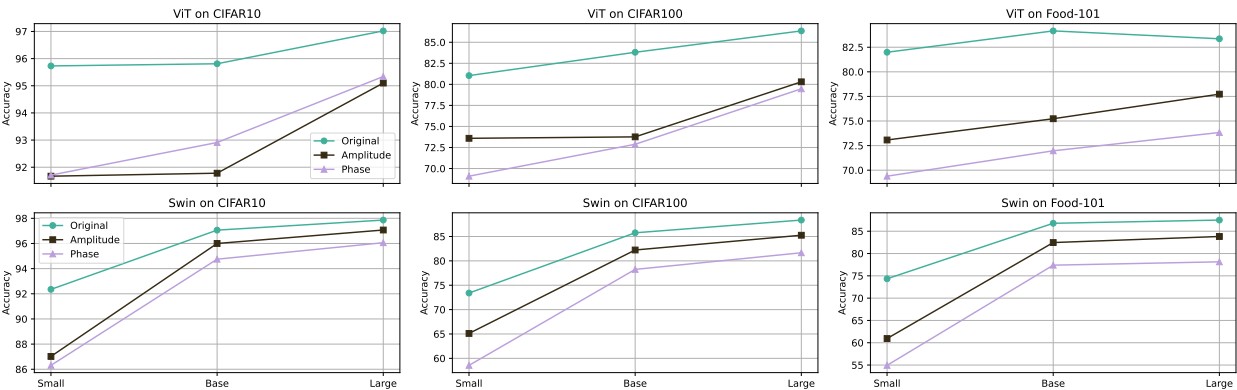

Figure 2: Classification accuracy of Vision Transformer (ViT) and Swin Transformer models of different scales (Small, Base, and Large) on CIFAR-10, CIFAR-100, and Food-101. Each plot compares classifiers trained on original pre-trained representations, their amplitude, and their phase.

Previous studies have also reported that CNNs favour amplitude over phase (Chen et al., 2021). In addition, the amplitude typically follows a power-law decay (Ruderman & Bialek, 1993; Simoncelli & Olshausen, 2001), which means that most of the energy is concentrated in the low-frequency components, resulting in a globally redundant structure. For example, in images of dogs, the low-frequency amplitude mainly captures the overall impression that "there is a large brown region with a light background." Because adjacent low-frequency components are highly correlated, amplitude patterns of images from the same class are almost repetitive. This redundancy makes the amplitude more robust to noisy labels and allows it to encode stronger generalisable information. In Figure 2, we further present results on two Transformer-based architectures—Vision Transformer (ViT) and Swin Transformer—across multiple model scales (Small, Base, Large). With the exception of ViT models on the low-resolution CIFAR-10 dataset, we observe a consistent trend where amplitude-based classifiers outperform their phase-based counterparts. This performance gap becomes more pronounced with larger model scales, higher-resolution datasets, and more fine-grained class distinctions. Given that CIFAR-10 is a relatively low-resolution dataset with limited semantic variation across its ten categories, its practical value is somewhat limited. Therefore, we argue that the observed superiority of amplitude features is likely to generalise well to real-world applications.

(2) In contrast, phase-based classifiers are more sensitive to noisy labels, resulting in poorer classification performance. The amount of class-discriminative information contained in the phase is relatively limited. Although positional and structural information of the image subject is indeed encoded in the phase, such geometric structures do not always directly reveal class distinctions, leading to relatively limited generaliza-

tion performance. On the other hand, in the phase domain, this local-focus bias causes small-scale shifts, which make the phase distribution more dispersed. This distorts the geometric information in the phase and further degrades classification.

(3) Second, on the three datasets—CIFAR-10, CIFAR-100, and Food-101, amplitude-based classifiers using 0%–10% noisy pre-trained representations show minimal performance differences, In fact, on CIFAR-100, there is even a slight improvement at the 10% noise level. This performance trend may be attributed to noisy labels inducing the pre-trained model to focus more on local features. Under mild noise, the model may start associating a label with unique details of a specific image (e.g., reflections or the edge of a particular pixel), and memorising additional textures may accidentally increase discriminability, explaining why the 5% noisy model in Chen et al. (2024) slightly outperforms the clean model in downstream tasks. Finally, on the ImageNet Val dataset, the performance of amplitude-based classifiers is nearly identical to classifiers using the original pre-trained features. This suggests that the pre-trained model we used indeed encodes most of the discriminative information into the amplitude.

Overall, from the classification results on the four image datasets, we find that the amplitude of pre-trained features encodes more label-related discriminative information, whereas the phase is more sensitive to noisy labels. They severely disrupt positional and structural information, leading to a more pronounced performance degradation for phase-based classifiers. To further investigate the impact of noisy labels on amplitude, we present the variation of the average amplitude magnitude of pre-trained representations from the CIFAR-10 dataset across different noise levels.

As shown in Figure 3, we observe that, as the noise level increases, we can clearly observe the gradual shift of magnitude in the feature amplitude from low frequencies toward mid- and high-frequency ranges, accompanied by progressively weakened structural patterns. Compared to clean pre-trained representations, the energy ratio in the low-frequency range (frequency index 0–200) decreases (from 46.98% to 45.50% to 44.37%) as noise increases (to 5% or 10%). In contrast, the ratios in the mid-frequency (200–600) and high-frequency (600–1024) ranges increase (mid frequency: from 25.76% to 26.42% to 27.40%; high frequency: from 27.20% to 28.08% to 28.23%) as noise increases. Under 5% noisy supervision, the first amplitude peak shifts from the low-frequency range to the mid-frequency range. The second peak, which originally appears around frequency index 400, is also delayed and emerges near index 600, while the changes in the high-frequency range remain minor. The spectral centroid increases from 2.18 to 2.48. For amplitudes obtained from 10% noisy supervision, both peaks are further shifted, with the first occurring around frequency index 400 and the second also around index 600.

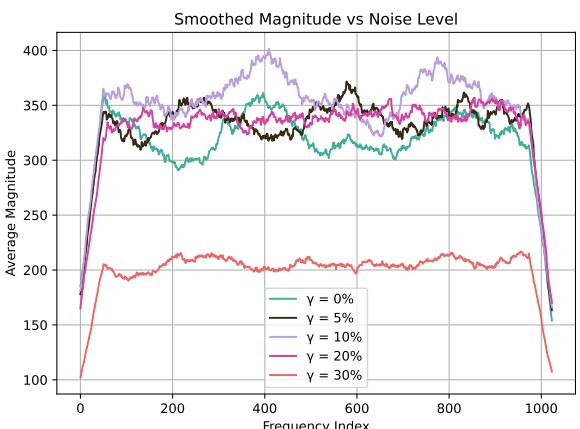

Figure 3: Smoothed amplitude magnitude across frequency indices under different noise ratios ($\gamma$). We divide the spectrum into three ranges based on the frequency index: low frequency (0–200), mid frequency (200–600), and high frequency (600–1000).

The spectral centroid continues to increase to 2.81. As the noise level increases, the normalized spectral entropy of representations also shows an increasing trend, from 74.98% to 77.31%, indicating a gradual weakening of structural regularity. At extreme noise ratios of 20% and 30%, the structure becomes disrupted, with no distinct peaks. At the 30% noise level in particular, the overall magnitude level is even reduced, indicating severe loss of discriminative information. Overall, we argue that the impact of noisy labels does not always manifest first as a simple high-frequency peak (Huang et al., 2023), but rather as a global alteration of the amplitude magnitude structure. Therefore, our correction method does not need to target any specific frequency band; instead, by leveraging amplitude residuals, it can more effectively strengthen or suppress particular frequencies as needed.

## 4 Amplitude-Guided Phase Realignment

**Motivation.** Through experiments presented in Section 4, we observe that: (1) discriminative information is primarily encoded in the amplitude, while the phase is highly sensitive to noisy labels; and (2) noisy labels in the pre-training dataset reshape both the distribution and the structural patterns of amplitude in pre-trained features. Since the distribution patterns of amplitude correspond to specific spatial structures, these structures in turn determine the distribution of phase. For example, repetitive textures (e.g., brick walls, grids) produce periodic peaks in the frequency domain, and the arrangement of phases at these frequencies decides the alignment of textures. Variations in amplitude also reflect changes in texture repetitiveness; if the periodic peaks are disrupted, it often indicates phase misalignment (Yu et al., 2022). Based on these observations, amplitude variations can serve as an effective indicator for guiding phase correction. The adjusted phase better aligns with the semantic content of the samples, eliminates the biased knowledge embedded in pre-trained features, and enhances generalisation on downstream tasks.

---

**Algorithm 1:** Amplitude-Guided Phase Realignment Classifier (Lorem)

**Input**         : Pre-trained features $\mathbf{F} \in \mathbb{R}^{B \times d}$
**Parameters:** MLP weights $(W_1, W_2)$ with BN+ReLU; classifier $W_c$; learnable template $\phi \in \mathbb{R}^{B \times d}$
                  (init $\pi$); step size $\varepsilon = 0.01$
**Output**       : Logits $\mathbf{M} \in \mathbb{R}^{B \times C}$

**Semantic branch:** $\mathbf{S} \leftarrow \mathrm{MLP}(\mathbf{F})$
**Fourier transforms (last dim):**
$\mathbf{X}' \leftarrow \mathcal{F}(\mathbf{F}), \quad \mathbf{S}' \leftarrow \mathcal{F}(\mathbf{S}) \; A_{\mathbf{X}} \leftarrow |\mathbf{X}'|, \quad \Phi_{\mathbf{X}} \leftarrow \angle(\mathbf{X'})$
$A_{\mathbf{S}} \leftarrow |\mathbf{S}'|, \quad \Phi_{\mathbf{S}} \leftarrow \angle(\mathbf{S}')$
**Amplitude residual:**
$r \leftarrow \tanh(A_{\mathbf{X}} - A_{\mathbf{S}})$
**Phase update:**
$\Delta\Phi \leftarrow r \odot \phi \quad \widehat{\Phi} \leftarrow \Phi_{\mathbf{S}} + \varepsilon\,\Delta\Phi$
**Reconstruction (inverse Fourier transform):**
$\hat{\mathbf{Z}} \leftarrow \Re\left( \mathcal{F}^{-1}\left(A_{\mathbf{S}} \odot e^{i\widehat{\Phi}}\right)\right)$
**Classification:**
$\mathbf{M} \leftarrow \hat{\mathbf{Z}} W_c^\top$
**return M**

---

### 4.1 Method

Our method aims to leverage amplitude residuals in the frequency domain to guide phase correction, thereby improving the generalisation of pre-trained representations on downstream tasks. Specifically, given an input batch $\mathbf{X}$, we extract the original pre-trained representation $\mathbf{F}$. We then transfer this representation into a new semantic space through a two-layer multi-layer perceptron (MLP) $h_\omega$, obtaining a new pre-trained representation batch $\mathbf{S}$. To enable comparison in the frequency domain, we apply the Fourier transform to both $\mathbf{F}$ and $\mathbf{S}$:

$$\mathbf{X}' = \mathcal{F}(\mathbf{F}) = A_{\mathbf{X}} \odot e^{i\Phi_{\mathbf{X}}}, \quad \mathbf{S}' = \mathcal{F}(\mathbf{S}) = A_{\mathbf{S}} \odot e^{i\Phi_{\mathbf{S}}}. \tag{1}$$

where $A_{\mathbf{X}}, A_{\mathbf{S}} \in \mathbb{R}^{B \times d}$ denote the amplitudes, and $\Phi_{\mathbf{X}}, \Phi_{\mathbf{S}} \in [-\pi, \pi]^{B \times d}$ denote the phases. Next, we compute the residual between the original and semantic amplitudes and regularise it with a $tanh(\cdot)$ function to ensure a stable training:

$$r = \tanh(A_{\mathbf{X}} - A_{\mathbf{S}}), \quad r \in (-1, 1)^{B \times d}. \tag{2}$$

Since the phase is an angular variable with values in $[-\pi, \pi]$, its update involves not only the magnitude of correction but also the sign (direction). However, the amplitude residual $r$ only reflects the strength of

the discrepancy between the two representations, lacking guidance on the direction of phase adjustment. To address this issue, we introduce a learnable phase template $\phi \in [-\pi, \pi]^{B \times d}$. By combining the residual $r$ with $\phi$, the model can adaptively learn which components require stronger correction and in which direction the correction should be applied. In other words, $\phi$ provides the phase correction with frequency selectivity, avoiding the under- or over-correction caused by uniform updates across all frequencies. Moreover, its learnability provides task adaptivity, enabling the model to automatically learn correction patterns that best fit a specific downstream task or data distribution, rather than relying solely on residual-driven updates. The phase correction term is given by the elementwise product of the amplitude residual $r$ and $\phi$, and can be represented as follows:

$$\Delta \Phi = r \odot \phi \qquad (3)$$

The updated phase is then computed as:

$$\hat{\Phi} = \Phi_{\mathbf{S}} + \varepsilon \cdot \Delta \Phi, \quad \hat{\Phi} \in \mathbb{R}^{B \times d} \qquad (4)$$

where $\varepsilon = 0.01$ is a global scaling factor that ensures gradual adjustments and stabilises training. We present the hyperparameter analysis in Appendix A.2. Here, we use $\Phi_S$ instead of $\Phi_X$ because $\Phi_X$ comes directly from pre-training under noisy supervision, and therefore inherently contains biased knowledge. If used directly, it would easily bring noise into the correction process.

Finally, the updated phase is combined with the semantic amplitude, and the inverse Fourier transform is applied to reconstruct the corrected representation:

$$\hat{\mathbf{Z}} = \Re\left(\mathcal{F}^{-1}\left(A_{\mathbf{S}} \odot e^{i\hat{\Phi}}\right)\right), \quad \hat{\mathbf{Z}} \in \mathbb{R}^{B \times d}. \qquad (5)$$

The operator $\Re(\cdot)$ denotes taking the real part to ensure that the reconstructed representation remains in the real space $\mathbb{R}^d$. The corrected representation $\hat{\mathbf{Z}}$ is then fed into a linear classifier $W_c$, and the entire model is optimised under the supervision of downstream labels. Ablation study on amplitude residual guidance and model designs are in Appendix A.1.

## 4.2 Discussion

One may argue that Lorem may fail if class-related texture patterns are not captured in the amplitude. To further investigate the effectiveness of our proposed method, we follow the experimental setup of Section 3 and design two types of OOD tasks: (1) Using the "real" subset of DomainNet as the training set and the "sketch" subset as the testing set, and vice versa; (2) Using the original CIFAR-10/100 training sets while applying four types of blur perturbations—Gaussian blur, motion blur, glass blur, and defocus blur—to the corresponding test sets.

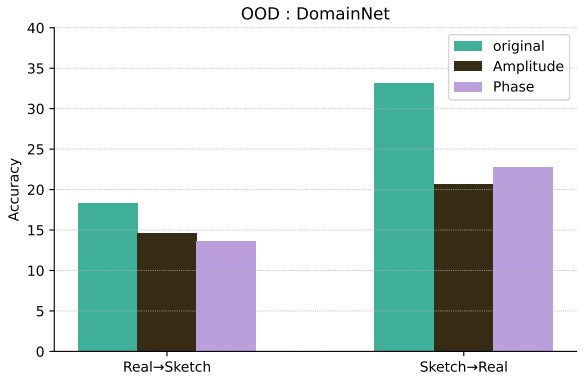

Figure 4: Classification accuracy on DomainNet using original pre-trained representations, their amplitude, and their phase when training on the "real" subset and testing on the "sketch" subset, and vice versa.

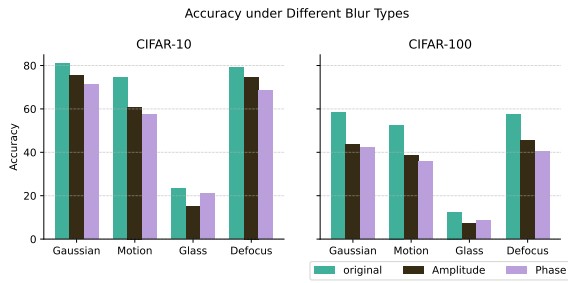

Figure 5: Classification accuracy on CIFAR-10/100 using original pre-trained representations, their amplitude, and their phase, when four types of blur perturbations are applied to the test sets.

As shown in Figure 4, when the "real" subset of DomainNet is used for training, the amplitude-based classifier slightly outperforms the phase-based classifier. However, when the roles are reversed, the phase-based classifier performs better than the amplitude-based one. This difference stems from the fact that the decision boundaries learned from the "real" subset rely heavily on texture information encoded in the amplitude. Although such information is removed in the "sketch" subset through decolorisation and texture removal, the overall shapes remain, thereby preserving transferability. When the training set becomes the "sketch" subset, which only contains positional and structural information, the amplitude-based classifier merely learns simple colour and texture patterns, causing it struggle when applied to the more complex "real" subset. Figure 5 shows the OOD results under the four blur types. The amplitude-based classifier outperforms the phase-based classifier in all but the glass blur setting, which swaps local pixel pairs, reassigning class-specific texture patterns, but leaves object edges and geometry unaffected; as a result, the phase retains some discriminative information. The other three types of blur mainly reduce fine-grained texture details while preserving the basic texture type, enabling amplitude-based classifier to remain effective.

In summary, our proposed method is able to achieve superior performance in most downstream tasks because the amplitude captures class-related texture patterns with relatively high generalisability. As long as such patterns are not completely destroyed in the target domain, the model can still extract relevant class-discriminative information.

## 5 Experiments

**Pre-trained models.** For vision pre-trained models, we select the fully supervised ResNet-50 (He et al., 2016) and Swin-L (Liu et al., 2021b), as well as the semi-supervised EfficientNet-B3 (Tan & Le, 2019). In addition, we include two noisy ResNet-50 pre-trained models trained on ImageNet-1K (IN-1K) (Russakovsky et al., 2015) with 5% and 10% label noise, provided by Chen et al. (2024). In their setup, noisy supervision is simulated by uniformly flipping ground-truth class labels into other classes. For text pre-trained models, we select BERT-Large (Devlin et al., 2019), RoBERTa-Large (Liu et al., 2019), and GPT-2 (Radford et al., 2019). Both BERT-Large and RoBERTa-Large are Transformer encoders, while GPT-2 is a Transformer decoder pre-trained on WebText using large-scale autoregressive language modelling.

**Datasets.** We validate our model on seven in-domain (ID) vision tasks and two out-of-domain (OOD) vision tasks. For the ID tasks, we use seven downstream datasets, including CIFAR-10/100 (Krizhevsky et al., 2009), Caltech101 (Fei-Fei et al., 2004), Food101 (Bossard et al., 2014), EuroSAT (Helber et al., 2019), RESISC45 (Cheng et al., 2017), and StanfordCars (Krause et al., 2013). For the OOD tasks, we use the "real" subset of DomainNet (Peng et al., 2019) as the training set and the "sketch" subset as the testing set, and vice versa. For text tasks, we validate our model on GLUE (Wang et al., 2018) and GLUE-X (Yang et al., 2022), for both ID and OOD evaluation.

**Baselines.** We compare our method with three related approaches: (1) Linear Probing: a single fully connected layer (Linear-Probe (FC)) and a two-layer MLP classifier (Linear-Probe (MLP)); (2) NMTune (Chen et al., 2024): a black-box method that applies singular value decomposition (SVD) to the original pre-trained features and introduces three regularisation strategies to adjust the Singular Value Entropy (SVE) and Largest Singular Value Ratio (LSVR), thereby transferring pre-trained representations into the new feature space and improving the generalisation of pre-trained models on downstream tasks.

**Experimental Setting.** We train each downstream classifier for 30 epochs using the Adam optimizer. The learning rate is set to 0.001 for the other baselines and 0.0001 for our proposed method. For evaluation, we report the average performance results over five runs.

### 5.1 Results

**Vision tasks.** Tables 1 - 2 show the results comparing our models to other baselines. We use the same datasets and identical train–validation–test splits, reporting the average accuracy across five runs on each dataset. Overall, our method consistently achieves higher accuracy in in-domain (ID) vision tasks, improving the quality of noisy pre-trained features. When using a single-layer linear network as the classifier, our approach significantly enhances the generalisation ability of pre-trained models on downstream tasks compared

Table 1: Performance comparison on ID tasks. Evaluation metrics: (1) "Acc" denotes accuracy; (2) "MPC" denotes mean per class. ●/○ indicates that Lorem is statistically better/worse than the compared method with the best performance by pairwised $t$-test at 95% confidence level.

| Model | Tuning | CIFAR-10 Acc | CIFAR-100 Acc | Caltech101 MPC | Food101 Acc | EuroSAT Acc | RESISC45 Acc | StanfordCars Acc | Avg |
|---|---|---|---|---|---|---|---|---|---|
| Resnet-50 | LP | $81.40 \pm 0.28$ | $59.60 \pm 0.42$ | $88.00 \pm 0.24$ ● | $49.95 \pm 1.54$ | $93.27 \pm 0.75$ | $79.37 \pm 1.09$ | $32.10 \pm 1.08$ | 69.10 |
| | MLP | $82.36 \pm 0.09$ | $59.07 \pm 0.44$ ● | $87.08 \pm 0.90$ | $55.96 \pm 0.24$ ● | $94.71 \pm 0.65$ | $84.28 \pm 0.26$ ● | $46.65 \pm 0.33$ ● | 72.87 |
| | NMTune | $81.37 \pm 0.27$ | $55.75 \pm 0.21$ | $84.98 \pm 0.51$ | $52.45 \pm 0.39$ | $94.74 \pm 0.21$ ● | $82.56 \pm 0.45$ | $42.78 \pm 0.20$ | 70.66 |
| | ours | $\mathbf{82.46 \pm 0.13}$ | $\mathbf{59.95 \pm 0.40}$ | $\mathbf{88.23 \pm 0.14}$ | $\mathbf{56.75 \pm 0.20}$ | $\mathbf{95.03 \pm 0.30}$ | $\mathbf{85.18 \pm 0.12}$ | $\mathbf{48.46 \pm 0.22}$ | 73.72 |
| EfficientNet-B3 | LP | $89.88 \pm 0.32$ | $69.55 \pm 0.31$ | $89.68 \pm 0.35$ | $67.52 \pm 0.18$ | $95.50 \pm 0.34$ | $86.77 \pm 0.20$ | $48.02 \pm 0.28$ | 78.13 |
| | MLP | $92.14 \pm 0.09$ | $72.53 \pm 0.22$ ● | $92.25 \pm 0.29$ ● | $\mathbf{72.18 \pm 0.13}$ | $96.53 \pm 0.05$ | $88.70 \pm 0.47$ ● | $57.85 \pm 1.98$ | 81.74 |
| | NMTune | $91.97 \pm 0.15$ | $70.22 \pm 0.49$ | $91.71 \pm 0.43$ | $69.57 \pm 0.28$ | $96.27 \pm 0.15$ | $87.43 \pm 0.55$ | $58.98 \pm 0.26$ ● | 80.88 |
| | ours | $\mathbf{92.19 \pm 0.08}$ | $\mathbf{72.85 \pm 0.14}$ | $\mathbf{92.63 \pm 0.33}$ | $72.11 \pm 0.23$ | $\mathbf{96.64 \pm 0.16}$ | $\mathbf{89.33 \pm 0.19}$ | $\mathbf{61.38 \pm 0.18}$ | 82.45 |
| Swin-L | LP | $95.68 \pm 0.24$ | $82.31 \pm 0.26$ | $93.83 \pm 0.32$ | $85.05 \pm 0.22$ | $96.53 \pm 0.39$ | $87.79 \pm 0.52$ | $50.78 \pm 1.99$ | 84.57 |
| | MLP | $96.42 \pm 0.12$ | $84.28 \pm 0.19$ ● | $95.72 \pm 0.17$ ● | $87.23 \pm 0.16$ | $97.23 \pm 0.39$ | $90.36 \pm 1.15$ | $64.19 \pm 1.99$ | 87.92 |
| | NMTune | $96.39 \pm 0.12$ | $82.67 \pm 0.23$ | $95.42 \pm 0.44$ | $85.69 \pm 0.10$ | $97.16 \pm 0.34$ | $89.38 \pm 0.27$ | $64.27 \pm 1.82$ ● | 87.28 |
| | ours | $\mathbf{96.48 \pm 0.08}$ | $\mathbf{84.45 \pm 0.17}$ | $\mathbf{96.18 \pm 0.32}$ | $\mathbf{87.26 \pm 0.11}$ | $\mathbf{97.33 \pm 0.17}$ | $\mathbf{90.67 \pm 0.46}$ | $\mathbf{69.72 \pm 0.21}$ | 88.87 |
| Resnet-50-5% | LP | $88.68 \pm 0.31$ | $68.76 \pm 0.28$ | $84.42 \pm 0.91$ | $57.41 \pm 0.45$ | $94.37 \pm 0.56$ | $83.84 \pm 0.41$ | $34.27 \pm 0.76$ | 73.11 |
| | MLP | $91.35 \pm 0.09$ | $72.17 \pm 0.08$ ● | $90.29 \pm 0.22$ ● | $\mathbf{63.19 \pm 0.19}$ | $95.96 \pm 0.34$ | $86.65 \pm 0.62$ ● | $49.11 \pm 0.52$ | 78.39 |
| | NMTune | $91.28 \pm 0.18$ | $69.87 \pm 0.34$ | $87.75 \pm 0.69$ | $60.09 \pm 0.11$ | $95.88 \pm 0.35$ | $85.68 \pm 0.27$ | $49.28 \pm 0.50$ ● | 77.12 |
| | ours | $\mathbf{91.42 \pm 0.19}$ | $\mathbf{72.54 \pm 0.33}$ | $\mathbf{91.00 \pm 0.35}$ | $63.14 \pm 0.28$ | $\mathbf{96.12 \pm 0.17}$ | $\mathbf{87.69 \pm 0.15}$ | $\mathbf{51.90 \pm 0.36}$ | 79.12 |
| Resnet-50-10% | LP | $88.28 \pm 0.26$ | $69.12 \pm 0.31$ | $83.38 \pm 1.13$ | $56.14 \pm 0.18$ | $94.59 \pm 0.14$ | $84.02 \pm 0.78$ | $34.34 \pm 1.24$ | 72.84 |
| | MLP | $91.14 \pm 0.09$ | $72.16 \pm 0.26$ | $87.60 \pm 0.35$ ● | $\mathbf{61.98 \pm 0.19}$ | $95.61 \pm 0.20$ | $87.04 \pm 0.43$ ● | $49.09 \pm 2.28$ | 77.80 |
| | NMTune | $90.96 \pm 0.22$ | $69.70 \pm 0.22$ | $85.51 \pm 0.84$ | $59.17 \pm 0.21$ | $\mathbf{95.88 \pm 0.31}$ | $85.89 \pm 0.58$ | $49.55 \pm 0.61$ ● | 76.67 |
| | ours | $\mathbf{91.30 \pm 0.14}$ | $\mathbf{72.37 \pm 0.22}$ | $\mathbf{87.99 \pm 0.27}$ | $61.83 \pm 0.21$ | $95.68 \pm 0.14$ | $\mathbf{88.01 \pm 0.41}$ | $\mathbf{52.39 \pm 0.13}$ | $\mathbf{78.51}$ |

Table 2: Performance comparison on OOD tasks. Evaluation metric: (1) "Acc" denotes accuracy. ●/○ indicates that Lorem is statistically better/worse than the compared method with the best performance by pairwised $t$-test at 95% confidence level.

| Model | Tuning | DomainNet Real Acc | DomainNet Sketch Acc | Avg |
|---|---|---|---|---|
| Resnet-50 | LP | $16.67 \pm 0.60$ | $31.08 \pm 0.37$ | 23.88 |
| | MLP | $19.69 \pm 0.21$ ● | $32.03 \pm 0.17$ | 25.86 |
| | NMTune | $17.61 \pm 0.31$ | $28.00 \pm 0.32$ | 22.81 |
| | ours | $\mathbf{20.02 \pm 0.14}$ | $\mathbf{32.19 \pm 0.36}$ | $\mathbf{26.10}$ |
| EfficientNet-B3 | LP | $23.57 \pm 0.15$ | $37.40 \pm 0.22$ | 30.49 |
| | MLP | $\mathbf{25.46 \pm 0.12}$ | $\mathbf{40.54 \pm 0.40}$ ○ | $\mathbf{33.00}$ |
| | NMTune | $23.03 \pm 0.20$ | $35.10 \pm 0.33$ | 29.07 |
| | ours | $\mathbf{25.46 \pm 0.09}$ | $37.70 \pm 0.26$ | 31.58 |
| Swin-L | LP | $38.67 \pm 0.29$ | $\mathbf{60.50 \pm 0.48}$ | 49.59 |
| | MLP | $\mathbf{40.38 \pm 0.41}$ | $59.65 \pm 0.87$ | $\mathbf{50.02}$ |
| | NMTune | $37.48 \pm 0.31$ | $58.64 \pm 0.27$ | 48.06 |
| | ours | $39.65 \pm 0.27$ | $60.34 \pm 0.73$ | 50.00 |
| Resnet-50-5% | LP | $18.02 \pm 0.33$ | $33.02 \pm 0.77$ | 25.52 |
| | MLP | $19.91 \pm 0.24$ | $\mathbf{36.68 \pm 0.29}$ | $\mathbf{28.30}$ |
| | NMTune | $18.20 \pm 0.13$ | $34.02 \pm 0.43$ | 26.11 |
| | ours | $\mathbf{19.94 \pm 0.15}$ | $36.22 \pm 0.08$ | 28.08 |
| Resnet-50-10% | LP | $17.64 \pm 0.12$ | $32.33 \pm 0.48$ | 24.99 |
| | MLP | $19.90 \pm 0.28$ | $\mathbf{36.05 \pm 0.22}$ ○ | $\mathbf{27.98}$ |
| | NMTune | $17.87 \pm 0.21$ | $33.30 \pm 0.34$ | 25.59 |
| | ours | $\mathbf{19.92 \pm 0.33}$ | $35.40 \pm 0.12$ | 27.66 |

to standard linear probing. In out-of-domain (OOD) vision tasks, the performance of our method is mixed. Nevertheless, the gap between our method and the best-performing approach is small, and it still outperforms other baselines. The reason for this discrepancy has already been analysed in the discussion section

Table 3: Performance comparison on GLUE tasks. Evaluation metrics: (1) "Acc" denotes accuracy; (2) "MCC" denotes matthews correlation; (3) "PCC" denotes pearson correlation. ●/∘ indicates that Lorem is statistically better/worse than the compared method with the best performance by pairwised $t$-test at 95% confidence level.

| Model | Tuning | CoLA MCC | MNLI Acc | MRPC Acc | QNLI Acc | QQP Acc | RTE Acc | SST Acc | STS PCC | Avg |
|---|---|---|---|---|---|---|---|---|---|---|
| BERT-L | LP | 38.97 ± 1.93 | **32.50 ± 2.33** | 70.06 ± 0.65 | 66.59 ± 3.31 | 77.07 ± 1.32 | **56.75 ± 1.32** | 83.53 ± 3.29 | **76.27 ± 0.94** | 62.72 |
| | MLP | 41.90 ± 1.03 • | 32.37 ± 1.80 | 67.93 ± 2.89 | **69.01 ± 0.57** | **84.18 ± 0.12** ∘ | 55.67 ± 2.01 | 85.64 ± 0.49 | 74.57 ± 1.23 | 63.91 |
| | NMTune | 40.11 ± 1.41 | 31.55 ± 1.80 | 69.10 ± 0.85 | 68.94 ± 0.60 | 83.80 ± 0.13 | 53.93 ± 1.22 | 86.31 ± 0.90 • | 76.16 ± 0.43 | 63.75 |
| | ours | 42.53 ± 0.91 | 32.50 ± 1.28 | 70.47 ± 1.21 | 68.12 ± 0.70 | 83.82 ± 0.14 | 55.67 ± 1.62 | **86.93 ± 0.96** | 75.43 ± 0.80 | 64.43 |
| RoBERTa-L | LP | 46.84 ± 1.19 • | 31.35 ± 3.20 | **72.07 ± 1.56** | 70.32 ± 2.70 | 78.37 ± 1.15 | 58.05 ± 3.19 • | **87.84 ± 0.26** | 66.78 ± 2.48 | 63.95 |
| | MLP | 45.46 ± 2.36 | 35.85 ± 3.42 • | 71.23 ± 0.44 | 70.64 ± 2.18 | 84.34 ± 0.59 | 54.37 ± 2.38 | 85.07 ± 4.02 | 68.96 ± 4.19 • | 64.49 |
| | NMTune | 39.15 ± 7.27 | 35.45 ± 5.23 | 69.26 ± 1.92 | 72.05 ± 0.79 | 84.48 ± 0.09 | 55.67 ± 1.56 | 86.95 ± 1.40 | 68.07 ± 3.52 | 63.88 |
| | ours | 48.19 ± 1.43 | 36.32 ± 2.86 | 71.32 ± 0.41 | **72.66 ± 0.99** | 84.64 ± 0.25 | 59.13 ± 1.77 | 86.63 ± 1.41 | **70.25 ± 3.45** | 66.14 |
| GPT-2 | LP | 17.19 ± 7.63 | 37.55 ± 22.25 | 70.41 ± 2.44 • | **67.24 ± 2.32** | 74.30 ± 0.57 | 50.04 ± 1.16 | **83.30 ± 0.30** | 44.99 ± 6.52 | 55.63 |
| | MLP | 20.15 ± 1.64 | 37.06 ± 22.13 | 69.63 ± 1.92 | 66.68 ± 1.55 | **82.43 ± 0.60** | 51.26 ± 1.84 • | 76.17 ± 3.71 | 58.43 ± 2.13 • | 57.73 |
| | NMTune | 16.31 ± 2.87 | 38.49 ± 17.02 | 66.72 ± 1.82 | 63.03 ± 3.71 | 80.91 ± 1.07 | 50.54 ± 1.29 | 58.06 ± 7.81 | 58.03 ± 2.32 | 54.01 |
| | ours | **22.22 ± 2.60** | **40.01 ± 17.46** | **71.59 ± 1.37** | 66.69 ± 1.37 | 80.70 ± 1.48 | **52.42 ± 1.15** | 81.36 ± 2.00 | **60.50 ± 1.59** | 59.44 |

Table 4: Performance comparison on GLUE-X tasks. Evaluation metrics: (1) "Acc" denotes accuracy; (2) "MCC" denotes matthews correlation. ●/∘ indicates that Lorem is statistically better/worse than the compared method with the best performance by pairwised $t$-test at 95% confidence level.

| Model | Tuning | GT MCC | IMDB Acc | MNLI-mis Acc | SNLI Acc | SICK Acc | NewsQA Acc | SciTail Acc | HANs Acc | Avg |
|---|---|---|---|---|---|---|---|---|---|---|
| BERT-L | LP | 42.64 ± 2.70 | 72.08 ± 6.29 | 52.77 ± 2.93 | 70.39 ± 3.61 | 53.36 ± 6.21 | 37.21 ± 3.64 | 59.16 ± 1.69 | 49.60 ± 0.36 | 54.65 |
| | MLP | 43.87 ± 1.60 | 72.12 ± 2.65 | 58.76 ± 0.44 | **74.39 ± 0.25** | 72.57 ± 3.54 | 37.27 ± 0.76 | 60.78 ± 1.48 | 48.69 ± 1.01 | 58.56 |
| | NMTune | 42.23 ± 2.03 | 72.63 ± 1.39 | 58.77 ± 0.69 • | 72.99 ± 0.71 | 75.92 ± 3.81 • | 37.41 ± 1.24 | 59.88 ± 1.46 | 48.59 ± 1.26 | 58.55 |
| | ours | **45.45 ± 3.88** | **72.69 ± 1.26** | **59.29 ± 0.59** | 74.38 ± 0.55 | **77.09 ± 2.97** | **38.83 ± 3.36** | **60.94 ± 1.16** | **50.07 ± 1.16** | 59.84 |
| RoBERTa-L | LP | 46.41 ± 2.73 • | **71.77 ± 2.23** | 74.09 ± 2.57 | 70.86 ± 2.29 | 43.98 ± 4.61 | 39.17 ± 3.45 | 55.47 ± 8.25 | **49.99 ± 0.01** | 56.47 |
| | MLP | 45.66 ± 11.52 | 59.45 ± 7.52 | 75.25 ± 2.63 | 71.80 ± 2.65 | 50.10 ± 3.61 | 38.95 ± 0.90 | 61.31 ± 8.04 • | 49.75 ± 0.29 | 56.53 |
| | NMTune | 39.81 ± 7.53 | 59.23 ± 5.59 | 76.35 ± 2.00 • | 72.67 ± 2.88 • | 50.83 ± 10.84 | 38.67 ± 1.84 | 60.43 ± 3.32 | 48.82 ± 0.85 | 55.85 |
| | ours | 47.54 ± 2.33 | 70.65 ± 4.75 | **77.40 ± 0.78** | 74.30 ± 1.13 | 57.59 ± 4.13 | 39.71 ± 1.87 | 62.61 ± 3.81 | 49.86 ± 0.11 | 59.96 |
| GPT-2 | LP | 41.02 ± 9.56 | 50.94 ± 0.69 | 58.43 ± 6.58 | 58.50 ± 2.30 | 62.45 ± 9.57 | 34.84 ± 0.39 | 55.75 ± 4.19 | 51.98 ± 0.87 | 51.74 |
| | MLP | 40.51 ± 16.09 | **51.05 ± 0.25** | 64.79 ± 0.84 | 63.32 ± 1.76 | 57.49 ± 3.10 | 37.72 ± 2.51 | 55.34 ± 7.45 | 52.92 ± 2.44 • | 52.89 |
| | NMTune | 42.48 ± 18.48 | 50.63 ± 0.28 | 57.88 ± 4.25 | 65.13 ± 0.78 | 57.30 ± 3.26 | 37.20 ± 2.64 | 41.51 ± 3.81 | 52.76 ± 1.36 | 50.61 |
| | ours | **43.66 ± 12.88** | 51.04 ± 0.16 | 63.76 ± 2.86 | 65.50 ± 1.10 | 62.87 ± 2.86 | 38.64 ± 3.61 | 58.11 ± 4.50 | 53.46 ± 1.09 | 54.63 |

(Section 4.2). Ablation study on amplitude residual guidance and model designs are in Appendix A.1. We also present the hyperparameter analysis in Appendix A.2.

**NLP tasks.** Tables 3 - 4 present the comparison of our method against other baselines on GLUE and GLUE-X tasks. GLUE and GLUE-X can be regarded as the ID and OOD tasks in NLP, respectively. Overall, our proposed method achieves the best average performance across both types of tasks, demonstrating its ability to adapt to diverse downstream tasks and dataset distributions, thereby improving the generalisation of pre-trained models. Another finding is that the MLP classifier also performs strongly in both ID and OOD tasks, ranking just below our method. In contrast, simple linear probing, while occasionally strong on specific tasks, shows unstable performance overall, consistently ranking last in average results. This conclusion differs from that of Chen's work, which we believe may be due to differences in learning rate selection.

## 6 Related Work

**Noisy Supervision.** Mainstream approaches can be broadly categorised into the following directions: (1) Noise Modelling (Van Rooyen et al., 2015; Han et al., 2018a; Van Rooyen & Williamson, 2018; Yao et al., 2019): Directly modelling the label noise generation process to correct the loss during feature learning; (2) Robust Loss Functions (Liu & Tao, 2015; Zhang & Sabuncu, 2018; Thulasidasan et al., 2019; Charoenphakdee et al., 2019; Lyu & Tsang, 2019; Menon et al., 2020): Modifying the optimisation objective to enable the model to learn features even under noisy labels; (3) Regularisation (Reed et al., 2014; Azadi et al., 2015; Zhang et al., 2018; Han et al., 2020): Introducing prior constraints so that the features are less sensitive to

noisy labels; and (4) Multi-model Learning (Veit et al., 2017; Li et al., 2017; Han et al., 2018b; Yu et al., 2019): Using multiple models to supervise each other, thereby reducing the risk of a single model being misled by noise. In contrast to these methods, our work focuses on mitigating the influence of biased knowledge in pre-trained representations caused by noisy labels. The most relevant to our work is Noisy Model Tuning (NMTune) proposed by Chen et al. (2024), a black-box fine-tuning method that employs multiple regularisation strategies. In Chen's work, the authors compare Singular Value Entropy (SVE) and Largest Singular Value Ratio (LSVR) computed from pre-trained features obtained under different levels of noisy supervision, analysing how noise affects representation learning and generalisation. As the noise ratio increases, LSVR rises and SVE becomes excessively high, indicating that the pre-trained features become more constrained to specific directions and less diverse on new data distributions—ultimately reducing transferability. Based on these observations, Chen proposes adjusting the pre-trained feature space to reduce the effect of noise and improve generalisation. We note that there has been relatively little work on understanding the influence of biased knowledge in pre-trained features. Our work approaches this problem from a frequency-domain perspective, examining how different frequency components change as noise levels increase. In contrast to Chen's findings, our conclusion is that higher noise levels cause pre-trained features to learn high-frequency details and unique texture patterns, rather than concentrating on a few specific dominant directions. Thus, our work offers a distinct interpretation of the problem from the frequency-domain viewpoint, providing new insights into this research area.

**Applications of Fourier Transform.** Frequency-domain analysis helps neural networks identify and preserve key features more effectively. Several studies have provided new insights into explaining the behaviour of neural networks from a frequency-domain perspective (Wang et al., 2020; Guo et al., 2020; Chen et al., 2021; Liu et al., 2021a; Yu et al., 2022; Zhou et al., 2024). In the visual domain, some research has found that high-frequency components play an important role in improving the accuracy of CNNs (Wang et al., 2020; Chen et al., 2021). In Chen et al. (2021), the authors observed that CNNs tend to converge to local optima closely related to the high-frequency components of training images. However, the high-frequency components are susceptible to noise and perturbations. Inspired by the phenomenon in the human visual system—where robust recognition relies more heavily on phase information—the authors proposed a novel data augmentation method that recombines the phase spectrum of the current image with the magnitude spectrum of a distractor image to generate new training samples. This approach encourages CNNs to focus more on structural information (derived from phase) and become more robust to variations in amplitude (such as noise, brightness, and colour distortions). In Huang et al. (2023), the authors investigated the differential impacts of phase and amplitude on CNN robustness, proposing a method to decouple phase and amplitude in certain layers via the Discrete Fourier Transform (DFT) during training, and applying distinct early-stopping strategies to each component, thereby enhancing the network's robustness. In natural language processing (NLP), research has shown that pre-trained large language models implicitly utilise Fourier features when performing arithmetic tasks (Zhou et al., 2024). MLP and attention layers leverage low-frequency and high-frequency Fourier components, respectively, to accomplish tasks, and different pre-training strategies directly influence the model's effective utilisation of these Fourier features. Our work also investigates the impact of noisy labels on pre-trained features from a frequency-domain perspective. The key distinction from prior work is that we focus on leveraging the observed characteristics to improve generalisation in downstream tasks, rather than modifying the pre-trained model itself or improving its training process.

## 7 Conclusion

In this paper, we propose a novel black-box method to mitigate the impact of noisy labels on the downstream performance of pre-trained models. The method leverages amplitude residuals to realign the original phase of pre-trained representations, thereby mitigating biased knowledge and improving generalisation. By analysing the performance of amplitude and phase extracted from pre-trained representations trained under varying levels of noisy supervision, as well as the changes in amplitude distribution induced by noisy labels, we gain deeper insights into how noisy labels distort phase, drive models to overfit on irrelevant texture patterns, and consequently preserve biased knowledge in pre-trained representations. Our algorithm employs amplitude residuals as guidance for phase correction, making the representations more robust and generalis-

able. Experimental results demonstrate that our method outperforms state-of-the-art baseline methods and widely used fine-tuning approaches. We conduct experiments on a variety of vision and language pre-trained models, and our method achieves competitive results on both in-domain (ID) and out-of-domain (OOD) tasks. It is important to note that our approach operates in the frequency domain of pre-trained representations. Whether it can be extended to large language models and other foundation models remains an open challenge, which warrants further exploration in future research.

### Acknowledgments

This work is partially supported by the Australian Research Council (ARC) Training Centre for Information Resilience (Grant No. IC200100022) and by an ARC Future Fellowship Project (Grant No. FT240100022). We acknowledge the authors of the paper "Understanding and Mitigating the Label Noise in Pre-training on Downstream Tasks" for making their pretrained models available, which greatly assisted our experimental analysis.

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

Table 5: Ablation study on amplitude residual guidance and learnable template across ID tasks. Evaluation metrics: (1) "Acc" denotes accuracy; (2) "MPC" denotes mean per class.

| Model | Variant | CIFAR-10 Acc | CIFAR-100 Acc | Caltech101 MPC | Food101 Acc | EuroSAT Acc | RESISC45 Acc | StanfordCars Acc | Avg |
|---|---|---|---|---|---|---|---|---|---|
| Resnet-50 | Amplitude guidance | $81.24 \pm 0.69$ | $59.26 \pm 0.51$ | $87.14 \pm 0.39$ | $56.51 \pm 0.40$ | $94.20 \pm 0.42$ | $83.59 \pm 1.39$ | $48.31 \pm 0.39$ | 72.89 |
| | Random phase shift | $81.11 \pm 0.94$ | $59.39 \pm 0.42$ | $87.44 \pm 0.41$ | $56.61 \pm 0.24$ | $94.76 \pm 0.43$ | $84.64 \pm 0.72$ | $48.15 \pm 0.25$ | 73.16 |
| | Fixed phase shift | $82.24 \pm 0.35$ | $59.13 \pm 0.46$ | $87.33 \pm 0.49$ | $56.44 \pm 0.30$ | $94.39 \pm 0.38$ | $84.24 \pm 0.57$ | $48.26 \pm 0.37$ | 73.15 |
| | Fixed template | $81.44 \pm 0.68$ | $59.46 \pm 0.37$ | $88.72 \pm 0.21$ | $56.63 \pm 0.24$ | $94.33 \pm 0.48$ | $84.19 \pm 0.40$ | $48.14 \pm 0.53$ | 73.27 |
| | ours | $82.46 \pm 0.13$ | $59.95 \pm 0.40$ | $88.23 \pm 0.14$ | $56.75 \pm 0.20$ | $95.03 \pm 0.30$ | $85.18 \pm 0.12$ | $48.46 \pm 0.22$ | 73.72 |
| EfficientNet-B3 | Amplitude guidance | $91.68 \pm 0.25$ | $71.29 \pm 0.23$ | $91.19 \pm 0.42$ | $65.28 \pm 0.17$ | $94.86 \pm 0.06$ | $86.38 \pm 0.58$ | $51.41 \pm 0.49$ | 78.87 |
| | Random phase shift | $91.39 \pm 0.22$ | $71.11 \pm 0.32$ | $90.43 \pm 0.74$ | $65.36 \pm 0.20$ | $94.96 \pm 0.21$ | $86.43 \pm 0.35$ | $51.41 \pm 0.24$ | 78.73 |
| | Fixed phase shift | $91.61 \pm 0.10$ | $71.34 \pm 0.24$ | $90.45 \pm 0.25$ | $65.57 \pm 0.18$ | $94.99 \pm 0.10$ | $86.39 \pm 0.62$ | $51.60 \pm 0.36$ | 78.85 |
| | Fixed template | $91.64 \pm 0.13$ | $71.22 \pm 0.40$ | $91.37 \pm 0.32$ | $65.60 \pm 0.08$ | $95.08 \pm 0.54$ | $86.07 \pm 0.46$ | $51.56 \pm 0.39$ | 78.93 |
| | ours | $92.19 \pm 0.08$ | $72.85 \pm 0.14$ | $92.63 \pm 0.33$ | $72.11 \pm 0.23$ | $96.64 \pm 0.16$ | $89.33 \pm 0.19$ | $61.38 \pm 0.18$ | 82.45 |
| Swin-L | Amplitude guidance | $96.40 \pm 0.07$ | $84.30 \pm 0.38$ | $94.84 \pm 0.18$ | $87.09 \pm 0.28$ | $97.28 \pm 0.17$ | $90.11 \pm 0.23$ | $69.37 \pm 0.49$ | 88.48 |
| | Random phase shift | $96.46 \pm 0.08$ | $83.91 \pm 0.32$ | $95.70 \pm 0.26$ | $87.15 \pm 0.23$ | $97.22 \pm 0.24$ | $90.38 \pm 0.56$ | $69.33 \pm 0.71$ | 88.59 |
| | Fixed phase shift | $96.48 \pm 0.11$ | $84.38 \pm 0.14$ | $95.86 \pm 0.13$ | $87.08 \pm 0.10$ | $97.18 \pm 0.19$ | $90.34 \pm 0.78$ | $69.11 \pm 0.66$ | 88.63 |
| | Fixed template | $96.39 \pm 0.03$ | $84.36 \pm 0.20$ | $94.68 \pm 0.12$ | $87.15 \pm 0.24$ | $97.12 \pm 0.38$ | $90.49 \pm 0.46$ | $69.58 \pm 0.77$ | 88.54 |
| | ours | $96.48 \pm 0.08$ | $84.45 \pm 0.17$ | $96.18 \pm 0.32$ | $87.26 \pm 0.11$ | $97.33 \pm 0.17$ | $90.67 \pm 0.46$ | $69.72 \pm 0.21$ | 88.87 |

# A  Appendix

## A.1  Ablation Study

The ablation study of Lorem is presented here, where we evaluate on the in-distribution (ID) vision datasets. We use three pre-trained vision models—ResNet-50, EfficientNet-B3, and Swin-L—for this analysis. We investigate the effectiveness of amplitude residual guidance and the learnable template. Specifically, we report results under different model configurations: (1) removing amplitude guidance; (2) applying a random phase shift; (3) using a fixed phase shift; and (4) using a fixed phase template, as shown in Table 5. From the results, one can observe that our proposed full model framework indeed improves downstream performance and yields more stable outcomes. In particular, the proposed method significantly outperforms the model variant without amplitude residual guidance on downstream tasks. Moreover, our method achieves notably better results on downstream tasks using EfficientNet-B3 compared to other pre-trained models.

We also conduct an ablation study on model design. Specifically, we report results under different model configurations: (1) using the original phase; and (2) adjusting the semantic amplitude with the residuals, as shown in Table 6. From the results, one can observe that replacing the semantic phase with the original phase significantly impairs downstream performance. The proposed model design — which adopts the semantic phase and applies phase adjustment instead of combining it with semantic amplitude — consistently enhances both accuracy and stability across downstream tasks.

## A.2  Hyperparameter sensitivity

The hyperparameter sensitivity analysis of global scaling factor $\varepsilon$ in Lorem is presented here, where we evaluate on three representative in-distribution (ID) vision datasets, which are CIFAR-10, CIFAR-100 (Krizhevsky et al., 2009), and Food 101 (Bossard et al., 2014). We use three pre-trained vision models—ResNet-50, EfficientNet-B3, and Swin-L—for this analysis. From the results in the table, choosing $\varepsilon = 0.01$ yields the best and most stable downstream performance across all three pre-trained models.

Table 6: Ablation study on model designs across ID tasks. Evaluation metrics: (1) "Acc" denotes accuracy; (2) "MPC" denotes mean per class.

| Model | Variant | CIFAR-10 Acc | CIFAR-100 Acc | Caltech101 MPC | Food101 Acc | EuroSAT Acc | RESISC45 Acc | StanfordCars Acc | Avg Acc |
|---|---|---|---|---|---|---|---|---|---|
| Resnet-50 | Original phase | $82.03 \pm 0.22$ | $58.21 \pm 0.58$ | $86.18 \pm 0.32$ | $55.22 \pm 0.52$ | $94.75 \pm 0.44$ | $84.54 \pm 0.65$ | $37.23 \pm 0.32$ | 71.17 |
| | Semantic amplitude | $82.31 \pm 0.24$ | $59.25 \pm 0.64$ | $87.52 \pm 0.21$ | $56.54 \pm 0.22$ | $94.99 \pm 0.49$ | $84.84 \pm 0.56$ | $47.58 \pm 0.80$ | 73.29 |
| | ours | $82.46 \pm 0.13$ | $59.95 \pm 0.40$ | $88.23 \pm 0.14$ | $56.75 \pm 0.20$ | $95.03 \pm 0.30$ | $85.18 \pm 0.12$ | $48.46 \pm 0.22$ | 73.72 |
| EfficientNet-B3 | Original phase | $91.77 \pm 0.16$ | $68.54 \pm 0.28$ | $89.03 \pm 0.74$ | $63.33 \pm 0.56$ | $94.62 \pm 0.19$ | $84.65 \pm 0.44$ | $37.92 \pm 0.27$ | 75.69 |
| | Semantic amplitude | $91.55 \pm 0.22$ | $71.39 \pm 0.13$ | $90.55 \pm 0.32$ | $65.51 \pm 0.31$ | $94.87 \pm 0.58$ | $86.33 \pm 0.28$ | $51.39 \pm 0.16$ | 78.80 |
| | ours | $92.19 \pm 0.08$ | $72.85 \pm 0.14$ | $92.63 \pm 0.33$ | $72.11 \pm 0.23$ | $96.64 \pm 0.16$ | $89.33 \pm 0.19$ | $61.38 \pm 0.18$ | 82.45 |
| Swin-L | Original phase | $96.42 \pm 0.15$ | $83.25 \pm 0.18$ | $94.37 \pm 0.25$ | $86.37 \pm 0.24$ | $97.29 \pm 0.12$ | $90.55 \pm 0.61$ | $52.91 \pm 0.54$ | 85.88 |
| | Semantic amplitude | $96.43 \pm 0.13$ | $84.43 \pm 0.24$ | $95.83 \pm 0.28$ | $87.11 \pm 0.21$ | $97.30 \pm 0.10$ | $89.96 \pm 0.45$ | $69.43 \pm 0.67$ | 88.64 |
| | ours | $96.48 \pm 0.08$ | $84.45 \pm 0.17$ | $96.18 \pm 0.32$ | $87.26 \pm 0.11$ | $97.33 \pm 0.17$ | $90.67 \pm 0.46$ | $69.72 \pm 0.21$ | 88.87 |

Table 7: Hyperparameter sensitivity study on global scaling factor $\varepsilon$ across ID tasks. Evaluation metrics "Acc" denotes accuracy.

| Model | $\varepsilon$ | CIFAR-10 Acc | CIFAR-100 Acc | Food101 Acc |
|---|---|---|---|---|
| Resnet-50 | 0.1 | $81.58 \pm 0.76$ | $59.12 \pm 0.81$ | $56.39 \pm 0.33$ |
| | 0.01 | $81.39 \pm 0.40$ | $59.19 \pm 0.28$ | $56.54 \pm 0.33$ |
| | 0.001 | $82.46 \pm 0.13$ | $59.95 \pm 0.40$ | $56.75 \pm 0.20$ |
| | 0.0001 | $81.66 \pm 1.02$ | $59.68 \pm 0.57$ | $56.68 \pm 0.38$ |
| EfficientNet-B3 | 0.1 | $91.64 \pm 0.22$ | $70.95 \pm 0.30$ | $65.85 \pm 0.21$ |
| | 0.01 | $91.64 \pm 0.10$ | $71.49 \pm 0.14$ | $65.70 \pm 0.11$ |
| | 0.001 | $92.19 \pm 0.08$ | $72.85 \pm 0.14$ | $72.11 \pm 0.23$ |
| | 0.0001 | $91.56 \pm 0.25$ | $71.49 \pm 0.14$ | $65.71 \pm 0.32$ |
| Swin-L | 0.1 | $96.32 \pm 0.12$ | $84.43 \pm 0.08$ | $87.25 \pm 0.12$ |
| | 0.01 | $96.31 \pm 0.20$ | $84.39 \pm 0.19$ | $87.21 \pm 0.35$ |
| | 0.001 | $96.48 \pm 0.08$ | $84.45 \pm 0.17$ | $87.26 \pm 0.11$ |
| | 0.0001 | $96.31 \pm 0.17$ | $84.27 \pm 0.33$ | $87.20 \pm 0.18$ |

