# OpenReview forum: "Denoising Pretrained Black-box Models via Amplitude-Guided Phase Realignment"
_TMLR — Accepted by TMLR_

### Review · Reviewer_cjqC · 2025-10-04

**Summary Of Contributions:**

The paper proposes **Lorem**, a **black-box denoising method** that mitigates **biased knowledge** in **pre-trained models** caused by **noisy labels**. It operates in the frequency domain, using **amplitude residuals** to guide **phase realignment** of representations  without modifying model parameters or accessing training data. The approach is tested on vision and language pre-trained models across in-domain (ID) and out-of-domain (OOD) tasks, outperforming linear probing, MLP, and NMTune baselines.

**Audience:**

Yes

**Audience Explanation:**

Members of the TMLR community, especially those focused on representation quality, frequency analysis, and robustness under noisy supervision, would be interested in the findings, even if the contribution is more empirical and diagnostic than state-of-the-art in performance.

**Broader Impact Concerns:**

No broader concerns.

**Claims And Evidence:**

No

**Claims Explanation:**

1. The claim that amplitude carries more discriminative information and phase is more noise-sensitive is partially validated by experiments on **CIFAR-10/100**. This lacks broader generalization to large scale datasets like **ImageNet**.

2. The finding that noise alters amplitude **globally** rather than only in high frequencies is qualitatively shown, but lacks quantitative metrics to confirm the trend.

3. The paper shows that its method improves accuracy slightly over baselines, suggesting the amplitude-guided phase correction might help. However, it provides no ablation studies to verify this. Without comparisons to variants that remove or change the guidance step, we cannot confirm that the observed gains actually come from the proposed mechanism. Thus, the causal role of amplitude-guided phase correction remains unproven.

**Requested Changes:**

1. Please run the amplitude–phase decomposition on ImageNet-pretrained
    ResNet-50 features (penultimate layer), and if feasible repeat on a
    standard ViT (e.g., ViT-B/16) to check architecture generality.
    Report: (a) amplitude-only vs phase-only linear-probe accuracies on
    ImageNet val (and a small transfer set), and (b) quantitative
    spectral metrics such as low/mid/high frequency energy fractions,
    spectral entropy, and a peak-shift index across clean vs
    synthetically noisy (5–10%) checkpoints. This to establish, at
    ImageNet scale (and ideally for a transformer too), that
    discriminative information is predominantly reflected in amplitude
    while phase is more perturbed by label noise, using both accuracy
    and spectral statistics rather than descriptive plots alone.


 2. Please include ablations to verify that the reported gains truly
    come from **amplitude-guided phase correction** rather than other
    factors.

     -   **Remove amplitude guidance:** Apply phase updates without using amplitude residuals.

    -   **Random or fixed updates:** Replace guided updates with random or constant phase shifts (same step size) to test causality.

    -   **Template ablation:** Compare fixed vs learnable phase templates to show the importance of adaptivity.

    -   **Phase source:** Use the original phase (Φₓ) instead of the semantic one (Φₛ) to confirm the design choice.

    -   **Hyperparameter sensitivity:** Vary ε and test the effect of removing tanh regularization to show stability.

    -   **Amplitude-only control:** Combine semantic amplitude with original phase to show that phase realignment is necessary.

3. Please add robustness experiments to verify that the proposed frequency-domain corrections improve stability. Evaluate adversarial robustness (PGD-L∞/L2) and if possible common corruptions (ImageNet-C), and include frequency-restricted attacks to show whether Lorem better resists low- or high-frequency perturbations. These tests will confirm that the claimed frequency alignment also enhances robustness under noise and distribution shifts.

4. Please report

    - Error & variability plots (statistical rigor)

    - Report 5-run mean ± std (already average is reported; add dispersion).

    - Confidence intervals (95%) or paired t-tests vs strongest baseline per dataset.

---

> ### Author Response · Authors · 2025-11-06
> **Response to Reviewer cjqC**
>
> First, we sincerely thank you for your valuable suggestions, which have greatly strengthened the experimental section of our paper by adding more statistically supported results and making it more complete. The corresponding revisions have been incorporated into the new version of the paper and are highlighted in red.
>
> > 1. Please run the amplitude–phase decomposition on ImageNet-pretrained ResNet-50 features (penultimate layer), and if feasible repeat on a standard ViT (e.g., ViT-B/16) to check architecture generality. Report: (a) amplitude-only vs phase-only linear-probe accuracies on ImageNet val (and a small transfer set), and (b) quantitative spectral metrics such as low/mid/high frequency energy fractions, spectral entropy, and a peak-shift index across clean vs synthetically noisy (5–10%) checkpoints. This to establish, at ImageNet scale (and ideally for a transformer too), that discriminative information is predominantly reflected in amplitude while phase is more perturbed by label noise, using both accuracy and spectral statistics rather than descriptive plots alone.
>
> (a) Thank you for helping us strengthen the motivating experiments. We have added additional results on two larger datasets: Food-101 and ImageNet-Val. The following tables present the experimental results:
> |              | original acc (%)                  | amplitude acc (%)                 | phase acc (%)                     |
> |--------------|-----------------------------------|-----------------------------------|-----------------------------------|
> | Food 101     | [66.60, 66.53, 66.60, 65.38, 63.78] | [60.37, 59.95, 59.69, 58.79, 55.57] | [52.77, 52.09, 51.22, 50.83, 48.66] |
> | ImageNet Val | [75.79, 74.05, 73.13, 71.83, 69.09] | [76.10, 73.77, 73.12, 71.39, 68.91] | [74.00, 71.22, 70.06, 69.49, 65.47] |
>
> In the new version paper draft, we present the results with new plot in Figure 1.
>
> (b) We did not include pre-trained ViT models that were trained on noisy datasets. However, in Section 3, we provide results using pre-trained features from ViT models of different scales and Swin Transformer models on CIFAR-10/100 and Food-101. We observe a consistent trend where amplitude-based classifiers outperform their phase-based counterparts. This performance gap becomes more pronounced with larger model scales, higher-resolution datasets, and more fine-grained class distinctions. Therefore, we argue that the observed superiority of amplitude features is likely to generalise well to real-world applications.
>
> > 2. Please include ablations to verify that the reported gains truly come from amplitude-guided phase correction rather than other factors.
>
> Thank you for this suggestion. We provide the ablation study on Appendix A.1.  We evaluate on seven in-distribution (ID) vision datasets. We use three pre-trained vision models—ResNet-50, EfficientNet-B3, and Swin-L—for this analysis. Specifically, we report results under different model configurations: (1) removing amplitude guidance; (2) applying a random phase shift; (3) using a fixed phase shift; and (4) using a fixed phase template. From the results, one can observe that our proposed full model framework indeed improves downstream performance and yields more stable outcomes. In particular, the proposed method significantly outperforms the model variant without amplitude residual guidance on downstream tasks. Moreover, our method achieves notably better results on downstream tasks using EfficientNet-B3 compared to other pre-trained models.
>
> We also conduct an ablation study on model design. Specifically, we report results under different model configurations: (1) using the original phase; and (2) adjusting the semantic amplitude with the residuals.
>
> > 3. Please add robustness experiments to verify that the proposed frequency-domain corrections improve stability. Evaluate adversarial robustness (PGD-L∞/L2) and if ...... distribution shifts.
>
> We appreciate the reviewer’s suggestion regarding robustness evaluation. In our current work, we have already conducted extensive OOD experiments and discussed in detail the importance of amplitude integrity in ensuring the robustness of our method under distribution shifts (Section 4.2). We believe that adversarial robustness, such as PGD-based attacks, targets a different aspect of model vulnerability and is not the main focus of this study. Instead, our OOD tasks are specifically designed to evaluate the stability of the proposed frequency-domain correction under realistic distribution shifts.
>
> > 4. Please report ......  paired t-tests vs strongest baseline per dataset.
>
> We provide the 5-run mean ± standard deviation and pairwise t-test results at the 95% confidence level. However, due to space limitations, we did not include the error and variability plots in the paper.

---

> > ### Comment · Reviewer_cjqC · 2025-11-16
> > **Final Reviewer Comments**
> >
> > Thank you for the careful and thorough revision. The added experiments, ablations, and statistical analysis clearly strengthen the paper and address my key concerns. The frequency-based motivation is now better supported, and the overall contribution is clearer and more convincing. I have updated my final recommendation accordingly.

---

> > > ### Author Response · Authors · 2025-11-19
> > > **Further Response to Reviewer cjqC**
> > >
> > > Thank you for your support of our work. The follow-up experiments you suggested have helped us further improve the study and make it more convincing.

---

### Review · Reviewer_bL7T · 2025-10-06

**Summary Of Contributions:**

The paper addresses the problem of biased knowledge in pre-trained models caused by noisy labels in large-scale datasets. Existing noise-robust learning methods typically require modifying training data or model parameters, which is impractical for massive and inaccessible pre-trained models.
To overcome this, the authors propose a black-box denoising framework, named "Lorem", which operates directly on pre-trained representations without retraining. The method analyzes feature representations in the frequency domain, finding that amplitude components encode most discriminative information while phase components are more sensitive to label noise.
Based on this insight, Lorem uses amplitude residuals to guide phase realignment, effectively removing biased knowledge.
Experiments on multiple pre-trained vision and language models demonstrate that this approach consistently improves downstream performance across both in-domain and out-of-domain tasks.

Strengths:

1. The paper introduces an amplitude-guided phase realignment method ("Lorem") for mitigating biased knowledge in pre-trained models without access to training data or model parameters, addressing a critical challenge in large-scale model adaptation.

2. The authors provide solid frequency-domain analysis that distinguishes between amplitude (discriminative) and phase (noise-sensitive) components, offering insightful theoretical grounding for their correction mechanism.

3. The proposed method is extensively validated on multiple pre-trained models (ResNet, Swin, EfficientNet, BERT, RoBERTa, GPT-2) and diverse in-domain (ID) and out-of-domain (OOD) tasks across vision and NLP, showing consistent gains.

4. The algorithm is computationally efficient and training-free for the backbone, making it practical for large-scale, frozen, or inaccessible models, with potential real-world usability.

**Audience:**

Yes

**Audience Explanation:**

Biased knowledge in pre-trained models caused by noisy labels in large-scale datasets is an important problem. The author provided an alternative method to address it.

**Broader Impact Concerns:**

No Broader Impact Concerns.

**Claims And Evidence:**

Yes

**Claims Explanation:**

The authors have conducted extensive experiments on vision and language tasks to show the effectiveness of their method in improving the pre-trained models trained on biased data in downstream tasks.

**Requested Changes:**

Though there are many strengths of the paper, there still remain some critical issues:

1. While empirically effective, the method lacks rigorous theoretical justification for the stability and convergence of the amplitude–phase correction process. The authors fail to present any proof to show why their method can yield effective result as they expected.

2. The improvement on OOD tasks (e.g., DomainNet in Table 2) is modest, and the analysis does not thoroughly dissect when and why performance gains diminish under certain domain shifts.

3. The approach assumes that discriminative information is primarily encoded in amplitude and that phase realignment universally improves generalisation. This assumption may not hold across all model architectures or modalities. As shown in Figure 1, only one set of vision experiments on CIFAR are shown. Besides, there is a significant drop in "amplitude" feature as well, while the proposed method keeps the amplitude in A_S without fixing the bias encoded in amplitude term.

4. Although the paper mentions potential extension to large language models, it provides no concrete experiments or adaptation strategy for such settings, leaving generalisation to larger-scale models speculative.

5. There are some minor issues:

A. Some typos exist, for example, empty parentheses in "different levels () of noisy", wrong equation in "X′ = F(F)" of Eqn. 1.

B. The full name of the proposed method Lorem is not specified.

---

> ### Author Response · Authors · 2025-11-06
> **Response to Reviewer bL7T**
>
> First, we sincerely thank you for your support of our paper and for the valuable suggestions you have provided. The corresponding revisions have been incorporated into the new version of the paper and are highlighted in red.
>
> > 1. While empirically effective, the method lacks rigorous theoretical justification for the stability and convergence of the amplitude–phase correction process. The authors fail to present any proof to show why their method can yield effective result as they expected.
>
> Our understanding is that you may be concerned about whether our correction method is theoretically stable. We can provide a proof showing that our method is Lipschitz-continuous. Our phase correction is :
>  \begin{equation}
> \hat{\Phi} = \Phi_\textbf{S} + \varepsilon \tanh(A_\textbf{X} - A_\textbf{S}) \cdot \phi
> \end{equation}
> We have:
> * $tanh(\cdot)$ is naturally bounded, with its output in the range (-1, 1);
> * $\varepsilon = 0.01$ is a small scaling factor;
> * $\phi \in [-\pi, \pi]^{B\times d}$ is a learnable phase template.
>
> Then, we have:
>  \begin{equation}
> \|\hat{\Phi} - \Phi_S\|_{\infty} \le \varepsilon \pi
> \end{equation}
>
> This represents a Lipschitz upper bound, indicating that our method ensures the norm of the phase update does not exceed $\varepsilon \pi$. We will include this in the Appendix section of the revised paper. Thank you for your suggestion — it will help strengthen the theoretical part of our work.
>
> > 2. The improvement on ....... shifts.
>
> We thank the reviewer for this observation. We would like to clarify that Section 4.2 (Discussion) already provides a detailed analysis of this behaviour.
>
> As discussed, the amplitude-based classifier relies on texture-related information captured in the low-frequency spectrum, which generalises well under appearance-level perturbations where overall shapes and coarse textures remain intact. However, its benefit diminishes in the “sketch” subset of DomainNet where colour and texture cues are removed entirely. In such cases, performance gains diminish when the domain shift disrupts amplitude-dominant texture cues but remain stable when only fine-grained appearance changes occur.
>
> In the revised paper, we further provide pairwise t-test results at the 95% confidence level, showing that our method’s performance on vision OOD tasks is not significantly weaker than the strongest baselines.
>
> > 3. The approach assumes that discriminative information is primarily encoded in amplitude and that phase realignment universally improves generalisation. This assumption may not hold across all model architectures or modalities. As shown in Figure 1, only one set of vision experiments on CIFAR are shown. Besides, there is a significant drop in "amplitude" feature as well, while the proposed method keeps the amplitude in A_S without fixing the bias encoded in amplitude term.
>
> We have added additional results on two larger datasets: Food-101 and ImageNet-Val. The following tables present the experimental results:
> |              | original acc (%)                  | amplitude acc (%)                 | phase acc (%)                     |
> |--------------|-----------------------------------|-----------------------------------|-----------------------------------|
> | Food 101     | [66.60, 66.53, 66.60, 65.38, 63.78] | [60.37, 59.95, 59.69, 58.79, 55.57] | [52.77, 52.09, 51.22, 50.83, 48.66] |
> | ImageNet Val | [75.79, 74.05, 73.13, 71.83, 69.09] | [76.10, 73.77, 73.12, 71.39, 68.91] | [74.00, 71.22, 70.06, 69.49, 65.47] |
>
> In the new version paper draft, we present the results with new plot in Figure 1. In Section 3, we provide results using pre-trained features from ViT models of different scales and Swin Transformer models on CIFAR-10/100 and Food-101. We observe a consistent trend where amplitude-based classifiers outperform their phase-based counterparts. This performance gap becomes more pronounced with larger model scales, higher-resolution datasets, and more fine-grained class distinctions. Therefore, we argue that the observed superiority of amplitude features is likely to generalise well to real-world applications.
>
> > 4. Although the paper ...... models speculative.
>
> Currently, the largest language model we experimented with is RoBERTa-L, which has approximately 355M parameters. We selected these three different language models because we aimed to demonstrate that, even under limited computational resources, our proposed method can still improve downstream task performance across different model architectures.
>
> > 5. There are some minor issues ...... specified.
>
> Thank you for carefully reviewing our paper. We have corrected this error in the revised version. In addition, Lorem is the name of a public playlist on a well-known music streaming platform that holds special meaning for me — I listen to it every day while working. I named my new model after this playlist as a small tribute to it.

---

### Review · Reviewer_WkqP · 2025-10-22

**Summary Of Contributions:**

The paper introduces “Lorem”, a black-box method for mitigating biased knowledge in pre-trained models caused by noisy labels, without requiring access to model parameters or training data. It operates purely on pre-trained representations by manipulating their frequency components—specifically the amplitude and phase derived via Fourier transforms.

**Audience:**

Yes

**Audience Explanation:**

the task is important as the growing size of pre-trained data

**Claims And Evidence:**

No

**Claims Explanation:**

- the method is based on the assumption that amplitude contains more discriminative information; however, it is not necessarily true since there is no theoretical justification; besides, some results in OOD tasks and noise types suggest the opposite and the validity in modern architecture such as ViT and larger dataset (ImageNet-21k) are not assessed.
- only experiments regarding ResNet is per-trained with noise, while other experiments is not clearly related to the claim that the proposed method could relieve the noisy pre-trained model for downstream tasks.
- the proposed Lorem is not convincing: basically, it tries recover the information loss caused by $g_{\theta}$ using amplitude containing more discriminative information; however, it does not imply the necessity of Fourier transforms as the original feature should possess the most information; besides, whether the improvement is caused by the proposal or increasing parameters $\phi$ is unclear; for example, what if we just simply connect $X$ and $S$ with a residual layer or concatenate them ?
- this work is based on pure empirical evidence; however, the current results is clearly not enough to support the claim compared to the related work [1], which provide substantial details of different experiment settings; for example, how would a simple regularization such as label smooth in pre-training, the size of $D'$ and the architecture of $g_{\theta}$ affect the final results is not fully investigated.

***
[1] UNDERSTANDING AND MITIGATING THE LABEL NOISE IN PRE-TRAINING ON DOWNSTREAM TASKS, ICLR 2024

**Requested Changes:**

see above

---

> ### Author Response · Authors · 2025-11-07
> **Response to Reviewer WkqP (1/2)**
>
> First, we sincerely thank you for your valuable suggestions. However, we also noticed several points that may have been misunderstood, and we would like to clarify them one by one below.
>
> > 1. the method is based on the assumption that amplitude contains ...... are not assessed.
>
> (a) We believe there may be a slight misunderstanding of our method’s motivation and focus. Our approach primarily emphasises how to perform phase correction, rather than exploiting the amplitude itself. In Section 3, we conduct two types of motivating experiments: one showing that classifiers using amplitude information achieve better downstream performance than those using phase information, and another showing that amplitude features are globally affected by different levels of noise. These results jointly demonstrate that (i) phase is the component that requires careful correction, and (ii) amplitude residuals serve as an effective indicator for guiding phase repair.
>
> In our work, we make an empirical observation supported by extensive experiments across diverse datasets and architectures (ResNet, ViT, Swin). Also, we include some theoretical explanations. The amplitude typically follows a power-law decay [1, 2] , which means that most of the energy is concentrated in the low-frequency components, resulting in a globally redundant structure. For example, in images of dogs, the low-frequency amplitude mainly captures the overall impression that “there is a large brown region with a light background.” Because adjacent low-frequency components are highly correlated, amplitude patterns of images from the same class are almost repetitive. This redundancy makes the amplitude more robust to noisy labels and allows it to encode stronger generalisable information.
>
> (b) We thank the reviewer for the observation regarding the OOD tasks on vision datasets. We would like to clarify that Section 4.2 (Discussion) already provides a detailed analysis of this behaviour. As discussed, amplitude-based classifiers rely on texture-related information captured in the low-frequency spectrum, which generalises well under appearance-level perturbations where overall shapes and coarse textures remain intact. However, their benefit diminishes in the sketch subset of DomainNet, where colour and texture cues are removed entirely. In such cases, performance gains diminish when the domain shift disrupts amplitude-dominant texture cues but remain stable when only fine-grained appearance changes occur.
> In the revised paper, we further provide pairwise t-test results at the 95% confidence level, showing that our method’s performance on vision OOD tasks is not significantly weaker than the strongest baselines.
>
> (c) Finally, our experimental setup already includes results using Swin Transformer (Large). We deliberately selected pre-trained models with different architectures to test the generality of our approach. In the revised version, Section 3 also includes results from ViT models of different scales and Swin Transformer variants, demonstrating that the observed superiority of amplitude-guided classifiers is consistent and likely to generalise to real-world applications.
> Although our computational resources do not allow us to include large-scale datasets such as ImageNet-21K, we have carefully designed our experiments to cover seven vision and eight language ID datasets, as well as two vision and eight language OOD datasets, ensuring that our conclusions are validated across diverse datasets and distributions.
>
> ---
> [1] Statistics of natural images: Scaling in the woods. Advances in neural information processing systems, 6, 1993.
> [2] Natural image statistics and neural representation. Annual review of neuroscience, 24(1):1193–1216, 2001.

---

> ### Author Response · Authors · 2025-11-07
> **Response to Reviewer WkqP (2/2)**
>
> > 2. only experiments regarding ...... tasks.
>
> We appreciate the reviewer’s comment and would like to clarify a possible misunderstanding.
> Our paper is based on the assumption that existing pre-training datasets inevitably contain a certain amount of label noise, meaning that any pre-trained model already carries some degree of biased knowledge in its parameters. Our method is a post-hoc correction applied to pre-trained representations, rather than a new pre-training procedure.
>
> Accordingly, in our experiments we selected several widely used pre-trained models in both vision and language domains, along with two models explicitly pre-trained under different noise levels, to evaluate how our method mitigates the effects of label noise on downstream tasks.
>
> > 3. the proposed Lorem is not ......  concatenate them ?
>
> We appreciate the reviewer’s thoughtful feedback and believe there might be a slight misunderstanding of our core idea.
>
> (a) Our method does not aim to “recover information loss” nor to directly use amplitude features. The amplitude and phase components are orthogonal; thus, simply using amplitude information would not make sense. Instead, we employ a learnable template whose range is $[-\pi, \pi]$, guided by amplitude residuals, which function more analogously to an attention mechanism—conducting phase correction based on global spectral consistency rather than replacing lost information.
>
> (b) To the best of our knowledge, there has been limited exploration of how noisy pre-trained representations behave in the frequency domain. The frequency domain, however, remains a powerful and popular lens for analysing visual representations. Our work complements the literature on noisy model learning by providing a frequency-based perspective.
> While prior studies often assumed that label noise primarily corrupts high-frequency components, our empirical findings suggest a more complex behaviour across the spectrum. Therefore, our contribution is not an “unnecessary transformation,” but rather a novel viewpoint. Moreover, our experimental results demonstrate consistent performance gains over multiple non-frequency-domain baselines.
>
> (c) In the revised version, we provide the ablation study on Appendix A.1. We evaluate on seven in-distribution (ID) vision datasets. We use three pre-trained vision models—ResNet-50, EfficientNet-B3, and Swin-L—for this analysis. Specifically, we report results under different model configurations: (1) removing amplitude guidance; (2) applying a random phase shift; (3) using a fixed phase shift; and (4) using a fixed phase template. We also conduct an ablation study on model design. Specifically, we report results under different model configurations: (1) using the original phase; and (2) adjusting the semantic amplitude with the residuals.
>
> These experiments confirm that our residual-guided phase correction mechanism is both necessary and effective—removing it leads to a notable decrease in downstream performance.
>
> > 4. this work is based on ...... fully investigated.
>
> (a) In the revised version of our paper, we have added substantial new experiments to strengthen the empirical evidence of our method: (1) classification results of amplitude-based and phase-based classifiers across four image datasets of different resolutions; (2) classification accuracy of Vision Transformer (ViT) and Swin Transformer models of different scales (Small, Base, and Large) on CIFAR-10, CIFAR-100, and Food-101; (3) analysis of smoothed amplitude magnitudes across frequency indices under varying noise ratios; (4) classification accuracy on the DomainNet dataset; (5) classification accuracy on CIFAR-10/100 under four types of blur perturbations; (6) comparison of our proposed method with four baselines across seven visual ID datasets and two OOD datasets; (7) comparison of our method with two baselines across seven language ID datasets and eight OOD datasets; (8) ablation studies on six different model variants (see Appendix A.1); and (9) hyperparameter sensitivity analysis (see Appendix A.2).
>
> If the reviewer still finds these results insufficient to demonstrate the rationality and effectiveness of our method, we would sincerely appreciate further suggestions.
>
> (b) We thank the reviewer for this comment. As mentioned, we have added detailed ablation studies in the appendix, evaluating six model variants to verify the necessity of our design.
> However, since our method is post-hoc, label smoothing, which is a regularisation technique applied during supervised training, is not applicable to our setting. Regarding different dataset scales, we currently lack sufficient computational resources to conduct additional tests. Nevertheless, we have carefully chosen pre-trained models of different scales and architectures to evaluate our method’s generality and practical effectiveness.

---

> > ### Comment · Reviewer_WkqP · 2025-11-17
> > **response**
> >
> > I appreciate the author's effort in revising the work. The new version, such as Fig. 2 indeed strengthens the motivation.
> > However, my __major concern__ remains unaddressed.
> >
> > - Judging from Fig.1,2, the original generally has the best performance, which raises a question about the necessity to use amplitude instead of original to revise phase, unless there is some proof that the amplitude-based features are more robust against noise compared to the original.
> > - Judging from the provided tables, MLP always gives at least the second best results (best in OOD), which raises a similar concerns regarding the necessity of Fourier based algorithmic design, where the improvement might come from increased parameters $\phi$. I notice that in the ablation study, the author shows the results of "Fixed template", which is related to this question; however, the comparison with MLP is not given.
> >
> > In addition, pre-trained features are denoted differently in text ($F$) and Algorithm 1 ($X$) .

---

> > > ### Author Response · Authors · 2025-11-18
> > > **Response to Reviewer WkqP -- Follow-up Comment**
> > >
> > > Thank you for the follow-up message. We address the major concerns point by point below.
> > >
> > > > 1. Judging from Fig.1,2, the original generally has the best performance, which raises a question about the necessity to use amplitude instead of original to revise phase, unless there is some proof that the amplitude-based features are more robust against noise compared to the original.
> > >
> > > Our understanding of your question is that you believe using the residuals of the pretrained features before the Fourier transform as a guidance signal to restore the phase is sufficient, and that the amplitude residuals are unnecessary. Our thoughts are as follows:
> > >
> > > (a) First, whether it is possible to use the residuals of pretrained features to restore the phase.
> > >
> > > We think this is not very reasonable, because phase exists in the frequency domain, whereas pretrained features are expressed in the spatial domain. They cannot be directly used unless an additional converter is introduced to map the pretrained feature residuals into the phase space. The problem with this approach is that, to repair the phase, one must already perform a Fourier transform to move into the frequency space, at which point using amplitude information becomes a more reasonable choice and avoids introducing an extra conversion module. Moreover, the residuals of pretrained features inherently contain both amplitude and phase residuals. Our assumption is that phase is highly affected by noise, so its residuals might not serve as a reliable guidance signal. In our paper, we argue that amplitude is at least more reliable because it may encode more stable and discriminative information learned by the model.
> > >
> > > (b) Second, whether pretrained feature residuals can be used to repair the pretrained features themselves.
> > >
> > > Suppose we have the residuals of pretrained features and use an MLP as a “residual block.” **We think this approach is essentially equivalent to fine-tuning.** Furthermore, converting features into the frequency domain and performing residual-guided repair there allows the proposed method to target "the corrupted parts" in a more directed way. If we simply use the pretrained feature residuals to repair the pretrained features themselves, it becomes difficult to explain what exactly is being corrected. Although we do not emphasize interpretability in our paper, our method does have potential in this aspect. Our approach focuses on the phase, and the residual used as a guidance signal can be decomposed into low, mid, and high-frequency components, which are particularly meaningful in vision tasks.
> > >
> > > > 2. Judging from the provided tables, MLP always gives at least the second best results (best in OOD), which raises a similar concerns regarding the necessity of Fourier based algorithmic design, where the improvement might come from increased parameters $\phi$. I notice that in the ablation study, the author shows the results of "Fixed template", which is related to this question; however, the comparison with MLP is not given.
> > >
> > > (a) We appreciate the reviewer’s careful reading of our ablation results. Indeed, removing the learnable phase template results in noticeably worse performance. However, we also observe that removing amplitude residual guidance (while keeping the learnable template) performs worse than using a fixed phase template combined with amplitude residual guidance.
> > > This strongly suggests that the performance gain comes from the amplitude-residual-guided mechanism, not from the template’s learnability alone.
> > >
> > > (b) Although the MLP fine-tuning baseline often achieves the second-best performance, our pairwise t-test results show that our method is statistically better than the MLP baseline on most vision tasks.
> > > For vision OOD tasks, our method does not always achieve the best performance, but only two cases are statistically worse than the MLP baseline.
> > > Overall, our approach shows a consistent advantage.
> > >
> > > (c) We have already included the MLP baseline in Table 1. Because of page limitations, the ablation results are placed in the Appendix, but the “ours” row in Table 1 corresponds directly to the proposed method compared against the MLP baseline.
> > > If the reviewer wishes to inspect the comparison in detail, Table 1 contains the necessary results.
> > >
> > > > 3. In addition, pre-trained features are denoted differently in text and Algorithm 1.
> > >
> > > Thank you for pointing this out. We have made the corresponding revisions and uploaded an updated version of the draft.

---

> > > > ### Comment · Reviewer_WkqP · 2025-11-19
> > > > **response**
> > > >
> > > > Thank you for the detailed discussion, which has addressed most of my concerns. As for an additional suggestion, the "Avg." column should be included in Tab. 5 & 6, which will make it easier for the readers to properly assess the contribution of each component. Once I have reviewed the update, I can recommend the paper for acceptance.

---

> > > > > ### Author Response · Authors · 2025-11-19
> > > > > **Further Response to Reviewer WkqP**
> > > > >
> > > > > Thank you for your support of our work. Your comments have helped us significantly refine and improve the paper. We have uploaded a revised draft that includes the added “Avg” column in Table 5 and Table 6.

---

### Decision · Action_Editor_4b5v · 2025-11-21

**Recommendation:** Accept as is

**Additional Comments:**

For this submission, the AE does not have any major concerns. However, as several reviewers have pointed out, the paper would benefit from a clearer and more substantial theoretical foundation, which could further strengthen the overall contribution.

**Audience:**

Yes

**Audience Explanation:**

The extensive experiments across multiple pretrained vision and language models demonstrate consistent improvements on both ID and OOD tasks, indicating that the proposed approach is practical and computationally efficient even for large models whose training data are inaccessible. Moreover, for researchers interested in representation quality, frequency-domain analysis, and robustness under noisy supervision, this work offers not only performance gains but also valuable empirical and diagnostic insights. Therefore, this paper is likely to be of interest to the TMLR audience.

**Claims And Evidence:**

Yes

**Claims Explanation:**

This paper introduces a black-box method called “Lorem,” which mitigates biased knowledge in pretrained models by performing amplitude-guided phase realignment without any retraining. In response to the reviewers’ concerns, the authors have carefully revised the manuscript, including additional experiments, ablations, and statistical analyses. Although the proposed method lacks strong theoretical justification, the reviewers judged that the authors’ claims are empirically supported by clear, accurate, and convincing evidence. The AE agrees with the reviewers’ assessment.